

# Morphology, Chemical Composition and Mixing State of Atmospheric Aerosols from Two Contrasting Environments in Southern India

Chandrika Rajendran Hariram[1], Gaurav Govardhan[1,2], Mohanan Remani Manoj[2], Narayana Sarma Anand[1,2], Karuppiah Kannan[3], Sreedharan Krishnakumari Satheesh[1,2], Krishnaswamy Krishna Moorthy[2]

[1]Divecha Centre for Climate Change, Indian Institute of Science, Bengaluru, India
[2]Centre for Atmospheric and Oceanic Sciences, Indian Institute of Science, Bengaluru, India
[3]Solid State and Structural Chemistry Unit, Indian Institute of Science, Bengaluru, India

*Correspondence to*: Chandrika Rajendran Hariram (hariramcrkattakada@gmail.com)

**Abstract.** The state of mixing of aerosols significantly influence their transformation, deposition, radiative forcing and health effects. We report the realistic (as in the atmosphere) state of mixing and morphology of aerosols at two nearby but contrasting environments; the urban region Bengaluru and a remote region Challakere. Ambient aerosols are collected on filter substrates using a High-Volume Sampler and analysed using Scanning Electron Microscope (SEM) equipped with Energy Dispersive X-Ray spectroscopy (EDX). The results show that the prevailing state of mixing of aerosols is 'core-shell' with $SiO_2$ as the core and Carbon-$SiO_2$-others (Calcium-Magnesium-Aluminium and other dust origin elements) combinations as the shell. On an average, for Bengaluru (Challakere), 66% (51%) of the core surface is coated. The sample-wise mean Carbon content of the composite particles reaches as high as ~26% (~9%) at Bengaluru (Challakere). The ambient black carbon (BC) mass concentration and the amount of rainfall that occurs just prior to the end of sampling are found to significantly influence the Carbon content of the particles. To the best of our knowledge, this is a first-of-its kind study over Indian region, coupling realistic aerosol observations and spectroscopy along with advanced image processing techniques to investigate the state of mixing and morphology of atmospheric aerosols at single particle resolution. This knowledge regarding the real-life state of mixing of aerosols would be useful in constraining the models studying aerosol-radiation interactions.

## 1 Introduction:

The knowledge of the state of mixing of the particles, which is the arrangement in which different aerosol species (dust, black carbon (BC), organic carbon, sulphate, sea-salts etc.) co-exist in the atmosphere, is reported to have great significance in the accurate determination of aerosol radiative forcing (RF) at the regional and seasonal scales (Jacobson, 2000;Chandra et al., 2004;Peng et al., 2016). Using realistic state of mixing (a core-shell model), Chandra et al. (2004) have demonstrated that incorporating the actual state of mixing of aerosols can remove some of the existing hypotheses in aerosol RF estimates, including the 'anomalous atmospheric absorption'. The mismatch between model-simulated and experimentally observed aerosol fields at regional and sub-regional scales is attributed to the uncertainties or inadequate representation of the state of mixing, in addition to other issues like inaccurate



boundary layer dynamics in the models, underestimated emissions of anthropogenic aerosol species (Nair et al., 2012;Moorthy et al., 2013;Govardhan et al., 2015;Govardhan et al., 2016) and trivialized humidity levels in the lower atmosphere (Pan et al., 2015;Feng et al., 2016). A few earlier studies (Jacobson, 2000, 2001;Lesins et al., 2002;Chandra et al., 2004;Schnaiter et al., 2005;Bond et al., 2006;Dey et al., 2008;Shiraiwa et al., 2008;Moffet and

Prather, 2009;Khalizov et al., 2009;Shiraiwa et al., 2010;Cappa et al., 2012;Ramachandran and Srivastava, 2016;Klingmüller et al., 2014;Wang et al., 2014;He et al., 2015;Liu et al., 2015;Peng et al., 2016;Chen et al., 2017) have clearly shown that the optical properties of aerosol species in-general and BC in-particular change substantially when their state of mixing changes from being purely external to core-shell or internal. It has been shown that the global mean RF of BC changes by 100% when it gets coated by all other aerosol species (Jacobson, 2000). Making

use of the Mie-scattering computations constrained by observed mass concentrations of different aerosol species, it has been reported that composite AOD shows an increment of ~ 60-80%, when BC coats sulphate aerosol or when BC is aged (Chandra et al., 2004). Bond et al. (2006) have shown that an enhancement of ~50% in BC absorption occurs when BC is coated by weakly absorbing aerosol species. However, many of these studies, while reporting enhancements in the optical properties of aerosols due to coating, have assumed the state of mixing of aerosols (BC

in particular) in absence of realistic observations of the same. Apart from modifying their optical properties, the mixing-state also influences the atmospheric life-time of aerosols (Van Poppel et al., 2005;Adachi and Buseck, 2008;Schwarz et al., 2008), their chemical properties, cloud interactions  (Tritscher et al., 2011;Cubison et al., 2008;Wang et al., 2010) and health effects (Stevanovic et al., 2013). All the aforementioned points highlight the importance of having better knowledge of the mixing-state of aerosols as they reside in the atmosphere. In the

backdrop of the above, we, in this study, report the results of examining the mixing state of aerosol particles from two close-by sites with contrasting anthropogenic influences; one a megacity – Bengaluru (BLR) and the other a remote rural environment - Challakere (CHK).

## 2 Methodology

### 2.1 Experimental Setup: Collection of Particles

Aerosol samples ($PM_{10}$) were collected on a quartz filter substrate using TISCH Environmental High Volume Sampler (Model: HIVOL+), within the campuses of the Indian Institute of Science, in Bengaluru (12.9° N, 77.6° E) and Challakere (14.35° N, 76.65° E); about 200 km apart (Fig. 1). The sampling locations represent contrasting environments. The Bengaluru campus of the Indian Institute of Science, though densely vegetated, represents background conditions for a highly polluted urban environment; being situated in the centre of the metropolitan city

of Bengaluru. On the other hand, the Challakere campus, located north-west of Bengaluru in Chitradurga district, is a representative of a semi-arid rural environment with flat-terrain and cleaner air due to far subdued anthropogenic activities. Over both the locations, the sampler was operated about 1.5 m above ground level, at a flow rate of 40 Cubic Feet per Minute (CFM), for the entire duration of sampling. Overall, 6 samples were collected at Bengaluru (BLR0 – BLR5) and 5 samples were collected at Challakere (CHK1 – CHK5) during the Indian summer monsoon

season. The sampling duration was kept around 6 – 8 hours for each sample. The entire sampling activity for each



station lasted for around 2 – 3 days. Separate samples were collected for polluted hours of morning and evening and relatively cleaner hours of afternoon. The sampling details namely the date, time of the day etc. can be found in Table 1. Subsequent to the collection of the particles, the samples were prepared for microscopic analysis following the guidelines outlined in Mukhopadhyay (2003) and Sielicki et al. (2011). Each of these samples were analysed using

Scanning Electron Microscope (SEM; FEI Siron XL30 FEG SEM and FEI Quanta 200 FEG SEM) equipped with EDX (Energy Dispersive X-ray spectroscopy). This technique has been widely used in the characterization of ambient aerosols (Casuccio et al., 2004;Haley et al., 2006;Kandler et al., 2007). The combined use of SEM and EDX revealed the morphology (shape and size) and elemental composition of each discrete particles. High-resolution surface images of individual particles were obtained using secondary electron detector (Everhat-Thornley solid state back scattered

electron detector) associated with the SEM at an accelerating voltage ranging from 5 kV to 20 kV. In our study, the cynosure was on particles having Equivalent Spherical Radius (ESR) less than 4 µm, keeping in mind their efficient interaction with the incoming solar radiation (Seinfeld and Pandis, 2016); their relatively longer atmospheric lifetime and their higher probability of causing respiratory problems (Brown et al., 2013) as compared to the bigger particles. Several images of such particles acquired from the samples at various magnifications ($2 \times 10^5$ to $2 \times 10^6$) were

subjected to careful, multiple area-wise EDX to estimate the average elemental composition. The EDX determines the characteristic elements, based on the energy levels of the X-rays collected from the area under focus. Around 400 selected particles from 11 different samples were examined using the aforementioned microscopic techniques. The number of particles analysed per sample can also be found in Table 1.

## 2.2 Post-processing of SEM Images:

As the raw SEM images alone were inadequate to infer detailed morphology of the collected particles, the images were post-processed through multiple image processing algorithms described below.

### 2.2.1 Contrast Enhancement:

The high accelerating voltage of the electron beam increases its penetration depth and diffusion area within the sample, which in-turn increases the noise and reduces the contrast in SEM images (JEOL, 2017;Sielicki et al., 2011). Hence,

to improve the contrast (and thus the clarity) within each SEM image, we used 'Contrast Limited Adaptive Histogram Equalization' (CLAHE) technique (Zuiderveld, 1994). The CLAHE, which is an extension of histogram equalization technique, improves the contrast within the image locally, by redistributing the pixel intensities to achieve more uniform distribution. This is achieved by controlling the redistribution of pixel intensities using the slope of the Cumulative Distribution Function (CDF) of the pixel intensity at that level. The CLAHE is widely used in image

processing applications (Reza, 2004;Chen et al., 2009;Yuan and Sun, 2012;Łoza et al., 2013). The application of CLAHE on a typical SEM image is shown in Fig. 2, where Fig. 2a is the original SEM image and Fig. 2b is the corresponding CLAHE enhanced image. It can be noticed that the CLAHE reveals the surface features within the SEM image more clearly.



### 2.2.2. Region Selection and Area Computation:

The SEM images with enhanced contrast were subsequently processed through 'Region Selection and Area Computation' (RSAC) image processing algorithm. This algorithm allows user to select a specific region within the contrast-enhanced SEM image. The selected region is cropped by binarizing the image (highlighting the selected region and zeroing down the pixel intensities of rest of the image). The area, perimeter and ESR of the selected region are further computed using the reference scale available in SEM images. This process of selection of region and area computation was carried out for each of the selected particles (400, in this study). The selection of the region is carried out manually, which is found to introduce a mean variation of 0.14% in the computed area, upon repeated selections. An example of region selection, cropping and area computation, from a SEM image can be found in Fig. 2c and Fig. 2d respectively.

### 2.3. Composition Retrieval:

The chemical composition of the selected particles was determined using EDX, which revealed the presence of Silicon (Si), Oxygen (O), Aluminium (Al), Calcium (Ca), Carbon (C), Iron (Fe), Sodium (Na), Magnesium (Mg), Potassium (K), and Chlorine (Cl); with traces of Titanium (Ti), Sulphur (S), Niobium (Nb), Manganese (Mn), Fluorine (F), Tin (Sn) and Phosphorous (P). A sample EDX carried out for 3 regions (a, b and c) within a selected particle is shown in Fig. 3. While the particle shows presence of C, Si, O and Ca with traces of Al, Mg and Cl, it clearly deciphers heterogeneous distribution of these elements.

### 3 Results and Discussion:

### 3.1 Broad Features:

The microscopic examination revealed that the aerosol particles existed in several different geometries including spherical, flat, agglomerate, irregular, cylindrical, crystal, ring, concave, chain and floccule; unlike the common assumption (spherical) in theory, but similar to those previously observed over different locations across the Indian region (Kumar et al., 2014;Pipal et al., 2014;Singh et al., 2014;Agnihotri et al., 2015;Tiwari et al., 2015;Pipal and Gursumeeran Satsangi, 2015;Mishra et al., 2015;Panda and Das, 2017). Such deviations of particles from spherical shape can alter the associated Single Scattering Albedo (SSA) (Mishra et al., 2015). It is further noted from the SEM images that, irrespective of the physical morphology, some of the collected particles were hollow, a few were porous, and a few were multi-layered. A closer examination revealed that most of them were coated with fine particulates. Most of the particles were found to have rough and uneven surfaces. SEM images of a few selected particles illustrating the aforementioned features are shown in Fig. 4. To compare and contrast the geometry of the particles collected at BLR and CHK, we broadly classify the collected particles into four sub-types, namely- flat and flat agglomerates (F&FA), spherical and spherical agglomerates (S&SA), chain and floccules (C&F), and miscellaneous (Misc). The last subtype (Misc) encompasses the particles which are crystal, ring, concave, cylindrical or hollow. The morphological distribution of the collected particles is illustrated in Fig. 5which reveals that the relative contribution of F&FA is higher over the rural station- CHK (75%) as compared to the urban station- BLR (45%). This is likely due



to the more availability of flat, irregular shaped mineral dust particles in the rural environment of CHK. The lesser presence of particles with C&F type geometry at CHK (12%) as compared to BLR (35%) could be linked with the far lesser ambient BC at CHK vis-a-vis BLR. This difference in ambient BC over the two stations could be partly due to the significantly higher number of automobiles plying in BLR compared to CHK. As on March 2017, the registered

number of vehicles in BLR are around 25 times more than that in the district of Chitradurga in which CHK is a taluk (TDGoK, 2016). The relationship between ambient BC and the state of mixing of aerosols will be explored later in section 3.4. In general, CHK reveals a clear dominance of F&FA over C&F, whereas they are roughly comparable in BLR.

### 3.2 Morphology of the Analyzed Particles:

General examination of the SEM images revealed that, most of the particles existed in a core-shell state of mixing, i.e., one or more species forming layers above the core surface, as shown in Fig. 4. The RSAC algorithm (section 2.2.2 and Fig. 2c-d) was used on the contrast enhanced SEM images obtained using the CLAHE technique (section 2.2.1 and Fig. 2b), to determine the projected area of the shell and the core regions of the particles. The core and the shell parts of the particles are manually identified. For example, in Fig. 2c, the regions within the blue line are identified as

the core, while the regions marked using red lines are identified as the shell. The ratio of projected area of the shell to that of the core was computed for each of the 400 particles. The mean of this ratio, calculated sample-wise for BLR and CHK, is given in Fig. 6. At both the locations, a partial coating of the parent or core material by the shell material was a common feature; however, the ratio of shell area to the core area differed significantly. For BLR (Fig. 6a), this ratio falls in the range from 0.63 to 0.71, with a mean of $0.66 \pm 0.01$, while for the cleaner environment of CHK, the

same ratio spanned from 0.43 to 0.58, with a mean of $0.51 \pm 0.03$ (Fig. 6b), with the coating percentage (i.e. the ratio of projected area of shell to that of the core) being ~23% lower than that in BLR. This enunciates the lesser availability (or lesser ambient concentration) of coating material (which in most cases is of anthropogenic origin, such as sulphate, carbonaceous etc) in the rural environment of CHK compared to the urban environment of BLR. This is explored and discussed further in section 3.4. Thus, our study has revealed that the ambient aerosol particles in urban as well as

rural environments exist in partial core-shell state of mixing; contrary to the assumptions made in the earlier studies (Chandra et al., 2004;Dey et al., 2008), where a fully coated core-shell model was considered.

### 3.3 Compositional Details of the Analyzed Particles:

Different portions of each of the selected particles were subjected to EDX analysis. The composition of each selected particle was computed by averaging these EDX results, and is, thus, a better representation of the entire exposed area

of the particle. The effective C coating percentage is estimated by excluding the carbonates and also by examining the typical chain structure of BC in SEM images. The sample-wise mean composition is shown in Fig.7 for BLR and CHK locations. It is seen that, on an average, for a randomly selected particle from BLR (Fig. 7a) and CHK (Fig. 7b) samples, Silicon and Oxygen dominate the chemical composition by contributing around 60-85%. Carbon is seen to be the third most abundant element in the composite particles, with sample-wise average contribution ranging from

3-26% (Fig. 7). Around 10-20% of the area of the particles is seen to be composed of other dust-origin elements like





Al, Mg, Fe, Mn, Sn, Ti, Ca etc. (Fig. 7). Merely 1-3% of the exposed area of the particles is composed of Na, Cl and S (Fig. 7). The subdued presence of Sulphur could be due to the evaporation of highly volatile sulphate aerosols in the high vacuum SEM environment (Ikegami et al., 1980;Ono et al., 1983;Deboudt et al., 2010). A recent study by Hamacher-Barth et al. (2016) reported that sulphate particles when exposed under SEM evaporated within a timescale
of a few minutes. It is further noted that:

    a)   For both BLR and CHK, the sum of C% and dust% (Si, oxides and other dust origin elements) across the collected samples remained almost constant.

    b)   The higher C share in the particles of BLR (Fig. 7a), compared to the relatively cleaner, rural ambiance of CHK with lesser combustion sources in the surroundings, corroborates the large contribution by the
10         automobile traffic, as discussed in section 3.1.

    c)   Significantly higher percentage of Si and oxides in the composite particles at CHK compared to that at BLR (Fig. 7) is attributed to the semi-arid nature of CHK, despite the extensive rainfall during monsoon.

A careful examination of the particle SEM images in the light of the EDX-retrieved composition reveals that:

    a)   The particles are mainly composed of $SiO_2$ (dust) in the core. These rough, irregular and porous dust particles
15         (as seen in Fig. 4) are found to be further coated by Carbon particles and smaller particulates of dust origin (Si, Mg, Al, Mn, Ca etc. and their oxides). This feature is common for both BLR and CHK.

    b)   However, due to the higher abundance of coating particles in BLR, the percentage of core area getting coated was much larger (66%) compared to that in CHK (~51%).

    c)   The sample-mean Carbon percentage reaches as high as 26% for BLR, while it remains around 9% for CHK.

The observed Carbon coating on dust is shown for two selected particles (Fig. 8) from BLR samples, along with their respective composition maps, generated with the help of SEM images and multiple EDX analysis. The figures 8a and 8c show original SEM images of the selected particles, while the figures 8b and 8d represent the corresponding compositional map. The black color in the map shows the presence of Carbon, while blue and red represents silica and other elements respectively. The composition maps clearly reveal partial coating of Carbon on dust cores.

**3.4 Relationship Between Carbon Coating and Ambient BC Mass Concentration: Effect of Rainfall**

It is obvious from the compositional details (Fig. 7) that the contribution of Carbon at CHK is very low (6.44% ± 1.01) compared to BLR (15.11% ± 2.92). To shed more light on this, we re-plot the sample-mean Carbon percentages for BLR (Fig. 9a) and CHK (Fig. 9b). For both the stations, it is observed that the Carbon percentage is not constant but varies within a relatively broad range for BLR and within a narrower range for CHK. For BLR (Fig. 9a), the samples
BLR0 and BLR5 have higher Carbon content (21% and 26%), while the other 4 samples (BLR1-4) depict around 10 – 12%. Similarly, for CHK (Fig. 9b) CHK2 and CHK5 depict higher contribution from Carbon (~9%) vis-a-vis the other samples (3 – 6%).To understand the causes behind such variations, we examined the ambient BC mass concentrations over the stations, measured continuously using an Aethalometer, (model AE-33 of Magee Scientific, USA). The Aethalometer measures ambient BC mass concentration by detecting change in the transmittance of a
quartz filter paper to an electromagnetic beam of wavelength 880 nm. More information about the working principle of Aethalometer may be found elsewhere (Hansen et al., 1984;Drinovec et al., 2015). The mean BC mass concentration



during the entire sampling duration is shown by a solid black line and their last 1-hour averaged values are shown using orange bars in Fig. 9c and Fig. 9d respectively. It is conspicuous that the ambient BC concentrations are relatively higher during the collection of samples BLR0 and BLR5, than that during BLR2, BLR3 and BLR4. This indicates that the ambient BC mass concentration has a strong say on the Carbon percentage in the composite particles

and also on the coating area. However, there are exceptions; for BLR1, the Carbon coating is around 10% (Fig. 9a) only, despite the higher BC mass concentration (Fig. 9c). Similarly, the Carbon content in CHK1 and CHK4 is less compared to that in CHK2 and CHK5, despite the prevailing higher ambient BC mass concentrations. To understand this, we examined rainfall measurements from a collocated disdrometer data, which provided rainfall every 5 minutes. For our analysis, we used the rainfall data during the last 1 hour of sampling (for CHK, collocated rainfall

measurements are not available and hence the TRMM daily rainfall data is used). BLR1 was collected during rainfall and the recorded rainfall for last 1 hour of sampling was more than 10 mm (Fig. 9e). Such high rainfall, though may not washout the ambient BC effectively due to the finer size of the BC particles(Seinfeld and Pandis, 2016), would affect the surface properties of the existing aerosols. The dust particles may become wet due to precipitation and high relative humidity. The hydrophobic nature of BC (Bond et al., 2013) may prevent them from coating over such wet

dust particles. Moreover, it is also possible that, the aged BC which coats over other species may get washed out due to rain, while the freshly emitted BC due to its hydrophobic nature may still stay afloat. These together could explain lesser Carbon percentage in BLR1 (Fig. 9a), despite the higher BC mass concentration (Fig. 9c). Similarly, on account of rainfall (Fig. 9e and Fig. 9f), BLR3, CHK1 and CHK4 also show lesser Carbon percentage compared to BLR2, CHK2 and CHK5 respectively, in spite of the higher BC mass concentration (Fig. 9c and Fig. 9d). Comprehensively,

it appears that the Carbon content in the analyzed particles has a direct relationship with the amount of BC available and this relationship is further controlled by the amount of rainfall, which could, to a certain extent, explain the lesser coating in CHK compared to BLR (Fig. 6).

To explore this relationship further, we have plotted Carbon percentage in the analyzed samples against ambient BC mass concentrations measured during the last one hour of sampling, for all the 11 samples (BLR0--5 and CHK1--5),

in Fig. 10. The color of the scatter-points in Fig. 10 signify the amount of rainfall during last one hour of sampling. It is evident that for all the dry samples (scatter points with zero rainfall), the Carbon percentage in the particles shows an expected linear relationship with the ambient BC mass concentration. The equation of the best-fit line (only for the dry samples) is shown in Fig. 10. The mean Carbon percentage in the particles is roughly 7 – 8 times of the available BC mass concentration in $\mu g\ m^{-3}$. This relationship gets disturbed during rainfall events. As seen previously, for the

same amount of ambient BC, rainfall reduces the amount of Carbon content in the sampled particles. This effect of rainfall on BC-Carbon content relationship is more obvious for higher amount of rainfall (e.g. for BC mass concentration around 3 $\mu g\ m^{-3}$, the Carbon content in the particles varies from 10 – 26% for rainfall ranging from 0 to 10.78 mm). Thus, we notice an expected co-variability in Carbon content in the sampled particles and the available BC mass concentration and an instrumental role played by rainfall in governing this relationship.

This study has quantified the shell-core state of mixing of BC and dust. The parent or core dust particles are seen to be partially coated by BC and smaller dust particulates. This type of state of mixing of carbonaceous and mineral dust aerosol species is not surprising for regions with high loading of both the species. A few previous studies have also



reported such a coating of Carbon on dust particles over the South Asian (Arimoto et al., 2006;Kumar et al., 2014) and the North-African (Deboudt et al., 2010) regions. Such a coating of carbonaceous species like BC on mineral dust would 'blacken' the dust and reduce its SSA making it more absorptive. The already recognized absorptive nature of Indian dust (Moorthy et al., 2007) could at least partly be related to such BC coating (though the iron enrichment

(Kumar and Sarin, 2009) might also contribute). This will be examined in detail in a subsequent work. Given the large impacts of the state of mixing of aerosols on the AOD (Chandra et al., 2004), these findings regarding ambient state of mixing of aerosol and its detailed morphology (shell area and core area), could serve as a critical input to the chemistry transport models in computation of AOD over the Indian region, which remains as an outstanding issue for such models (Pan et al., 2015;Govardhan et al., 2015;Govardhan et al., 2016). Furthermore, the BC aerosol species

are inherently classified into fine mode (with equivalent diameter < 1 μm) with atmospheric residence time of about a week (Ramachandran and Srivastava, 2016;Bond et al., 2013). However, when such BC coats the heavier aerosol species like dust (as revealed in this study), it would get removed from the atmosphere faster, due to the shorter atmospheric residence time for mineral dust species owing to its size and mass. Moreover, such depositions of BC on mineral dust aerosol, would also alter the health-related side-effects of the dust species. We notice a direct relationship

between Carbon content of the collected particles and the atmospheric BC mass concentration. In future, it would be interesting to examine whether the linear nature of this relationship holds for stations with higher BC mass concentrations. For this, aerosol sampling will have to be carried out at more polluted stations. The state of mixing of aerosols reported here pertains only to the summer monsoon season. It is possible that the mixing state of aerosol could be different in other seasons, that will be explored in a subsequent follow-up study. This study has revealed the

existence of blackened dust at near-surface levels over the Indian region. Such a deposition would reduce the SSA associated with the dust and enhance the consequent solar absorption. The resulting enhanced atmospheric warming would be more severe for middle-tropospheric heights than at the surface, due to the rarer atmosphere at those heights. This underlines the importance of sampling of aerosol particles from those middle tropospheric levels (4 – 5 km) and further characterization of the state of mixing of the collected particles from those levels. This would improve our

understanding on the columnar atmospheric warming due to aerosols.

## 4 Conclusions

The present study unravels the existing uncertainties regarding the morphology and mixing state of aerosols, hitherto unexplained, collected over two nearby, but contrasting environments (an urban centre, Bengaluru and a remote, rural location, Challakere, about 200 km apart) through a combination of SEM-EDX analysis and advanced image

processing techniques. The knowledge on the realistic state of mixing of aerosols is inevitable for the accurate estimation of radiation budget and aerosol-cloud interactions which can reduce the prevailing uncertainties hovering around aerosols. The important outcomes are:

1. Most of the particles in the rural location were flat shaped (~76%), while those in the urban centre were either flat (~46%) or chain shaped (~32%).

2. Irrespective of the shape (plates, floccule, rings and cylindrical), the particles were found to be layered, in 'core-shell' state of mixing.





3. In general, silica or $SiO_2$ (mineral dust origin) was seen to form the base or core of the particles and the combination of Carbon, $SiO_2$ and a few other elements (Ca, Al, Mg and traces of Na, Cl and S) formed the coating (shell).

4. Though only partial, the percentage of coating was higher in the urban environment. On an average, in the polluted (rural) ambiance of Bengaluru (Challakere), around 66% (51%) of the core area was found to be coated.

5. The sample-mean Carbon content within the collected composite particles reached as high as 26% in the urban location, while it was less than 10% in the dust dominant rural location.

6. The percentage of Carbon coating depicted a linear relationship with the ambient BC mass concentration during dry conditions. However, strong rainfall seems to vitiate this relationship, leading to lesser percentage of Carbon in the shell.

*Acknowledgements.* This work is partially supported by MoES (grant no. MM/NERC-MoES-1/2014/002) under the South West Asian Aerosol Monsoon Interactions (SWAAMI) project. We express our sincere thanks to Prof. G. S. Bhat, Centre for Atmospheric and Oceanic Sciences (CAOS), Indian Institute of Science (IISc), Bengaluru for providing the rainfall data. A part of this work was carried out at Micro and Nano Characterization Facility (MNCF) located at CeNSE, IISc, Bengaluru, funded by NPMAS-DRDO and MCIT, MeitY, Government of India. We would also like to acknowledge Advanced Facility for Microscopy and Microanalysis, IISc, Bengaluru. We extend our gratitude to Mr. Hanumantha Reddy G., Project Assistant, CAOS, IISc, Bengaluru for assisting in sample collection. Gaurav and Anand acknowledge the Grantham fellowship awarded by Divecha Centre for Climate Change. The authors would like to thank the TRMM mission scientists and associated NASA personnel responsible for the TMPA products.

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



| Sample name | Place | Date | Sampling time Start time-End time (IST hours) | No. of particles examined |
|---|---|---|---|---|
| BLR0 | Bengaluru | 08-08-16 | 18:00-23:00 | 21 |
| BLR1 | Bengaluru | 19-08-17 | 09.45-16.45 | 41 |
| BLR2 | Bengaluru | 19-08-17 | 22.30-06.30 | 47 |
| BLR3 | Bengaluru | 20-08-17 | 09.30-15.30 | 59 |
| BLR4 | Bengaluru | 20-08-17 | 19:00-03:00 | 37 |
| BLR5 | Bengaluru | 30-08-17 | 18:00-02:00 | 44 |
| CHK1 | Challakere | 16-08-17 | 14.00-19.00 | 34 |
| CHK2 | Challakere | 17-08-17 | 00:30-06:30 | 24 |
| CHK3 | Challakere | 17-08-17 | 09:00-15:00 | 26 |
| CHK4 | Challakere | 17-08-17 | 18.00-23.30 | 31 |
| CHK5 | Challakere | 18-08-17 | 03:00-10:00 | 36 |

**Table 1: Details (date, duration and number of particles analysed) about sampling carried out at Bengaluru (BLR) and Challakere (CHK). The samples BLR1, BLR3, CHK1 and CHK3 are collected during the cleaner hours of afternoon, while the remaining ones are collected during the relatively polluted hours.**





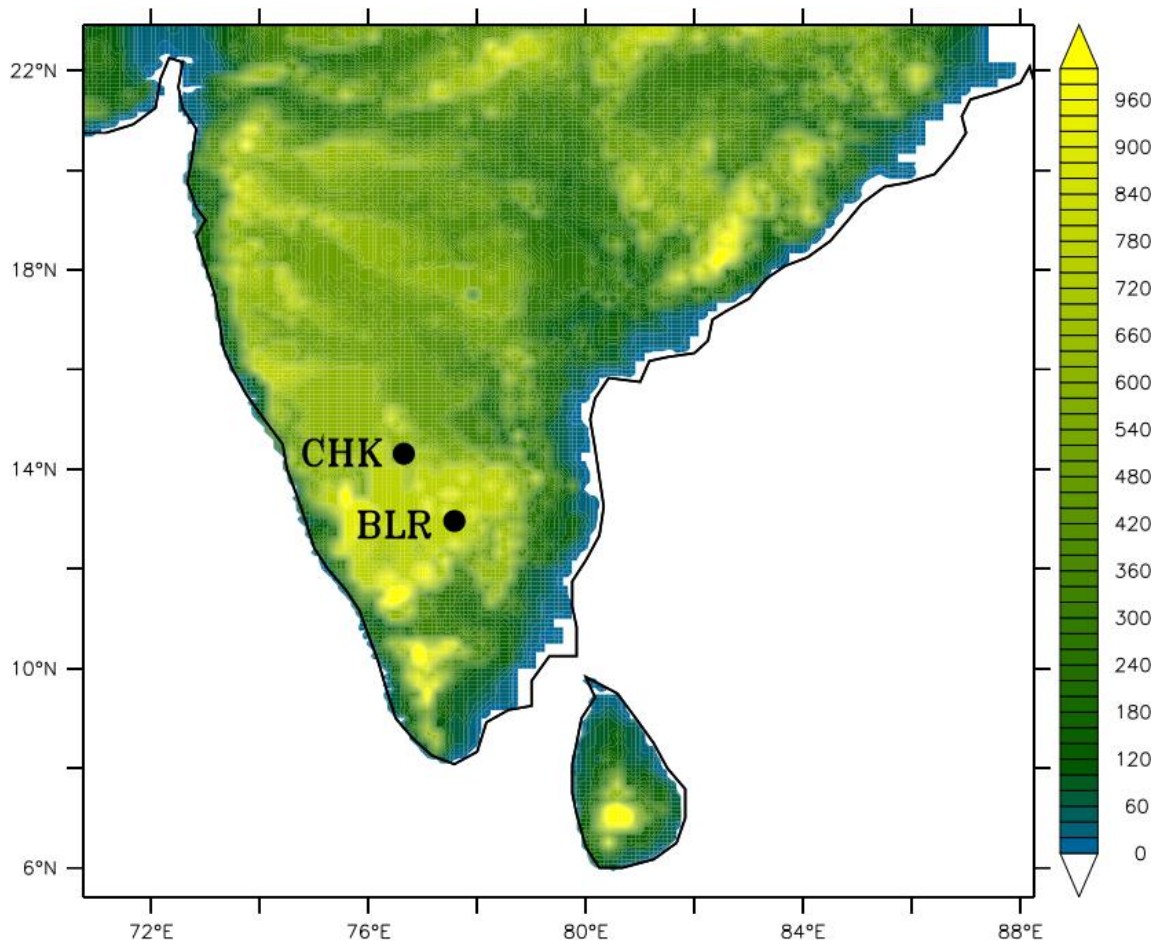

**Figure 1: The sampling stations (BLR- Bengaluru and CHK- Challakere) are marked with a dot on the map of India. The contours indicate surface elevation (m). The surface elevation has been plotted using ETOPO5 data of the National Centres for Environmental Information (NCEI) of National Oceanic and Atmospheric Administration (NOAA), with 5-minute resolution in latitude and longitude.**



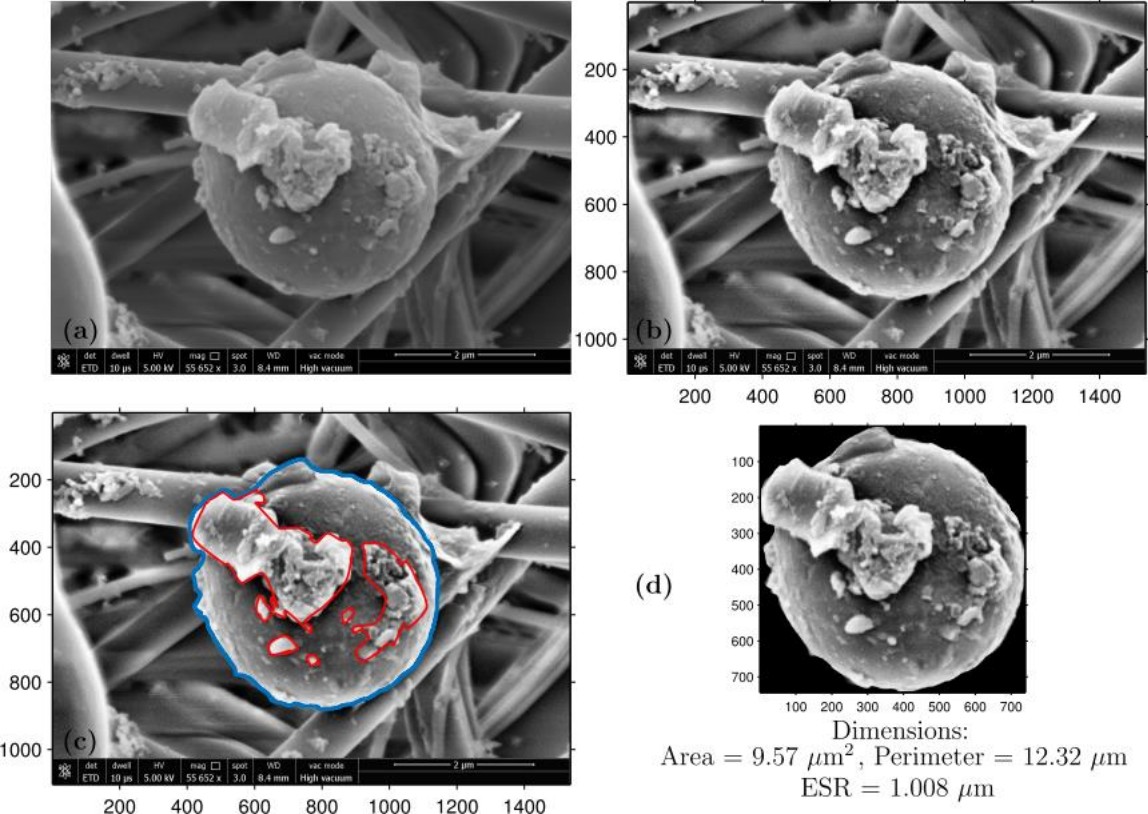

**Figure 2: a) original SEM image b) the image obtained after enhancement using CLAHE technique c) illustration of Region Selection algorithm on the particle; blue boundary for core and red boundaries for shell area selection d) image of the particle showing the selected region cropped from the background and dimensions of the selected region.**



25

**Figure 3: a) EDX spectrum of the region 'a' in the embedded SEM image b) EDX spectrum of region 'b' in the embedded SEM image c) EDX spectrum of region 'c' in the embedded SEM image.**

30





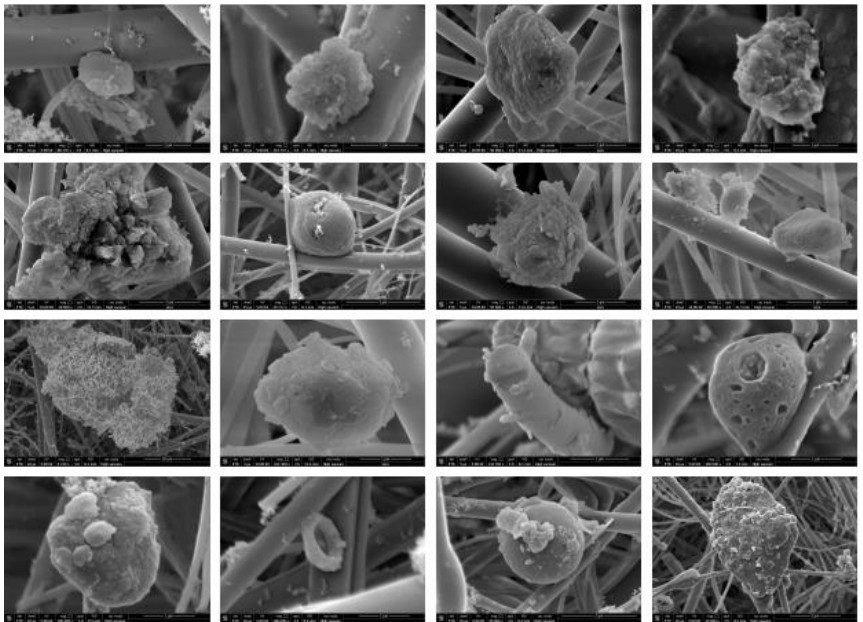

**Figure 4: SEM images of particulates illustrating broad morphological features.**



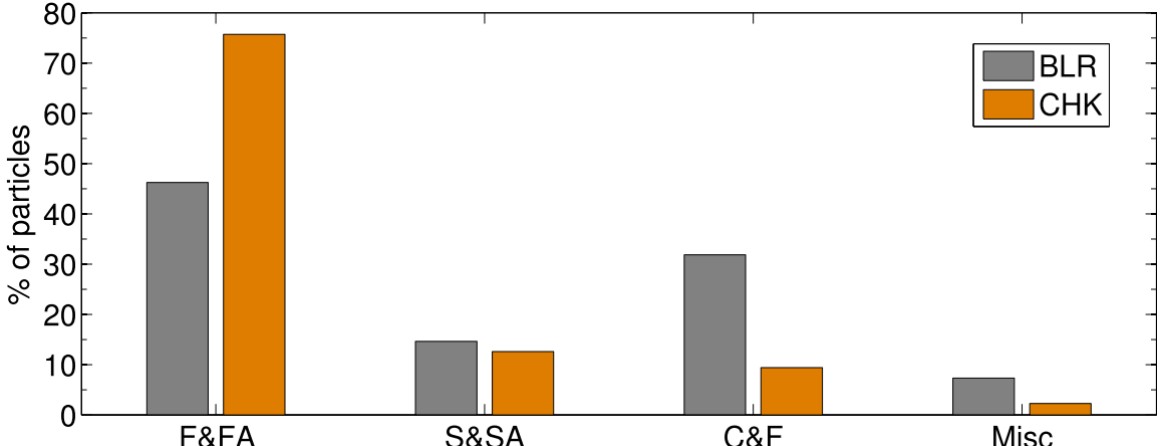

**Figure 5: Broad classification of the particles collected at BLR and CHK based on the geometry. The expansion of the acronyms are as follows: F&FA- Flat and flat agglomerate, S&SA- Spherical and spherical agglomerate, C&F- Chain and floccule, Misc- Miscellaneous (including crystal, ring, concave, cylindrical or hollow shaped particles).**




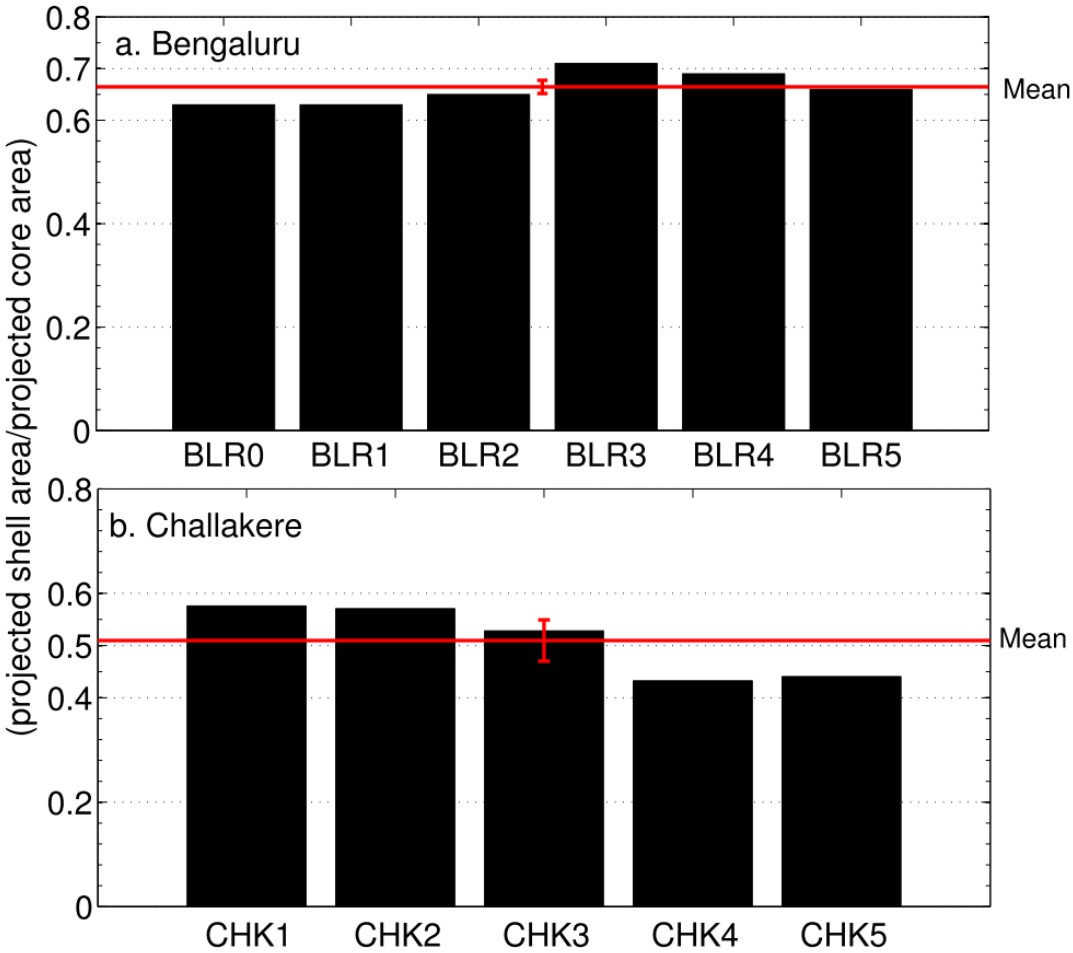

**Figure 6: Sample-wise mean ratio of projected area of shell to that of the core for a) Bengaluru and b) Challakere. The**
**station-wise mean has been indicated by a red line with an error bar depicting the standard error.**





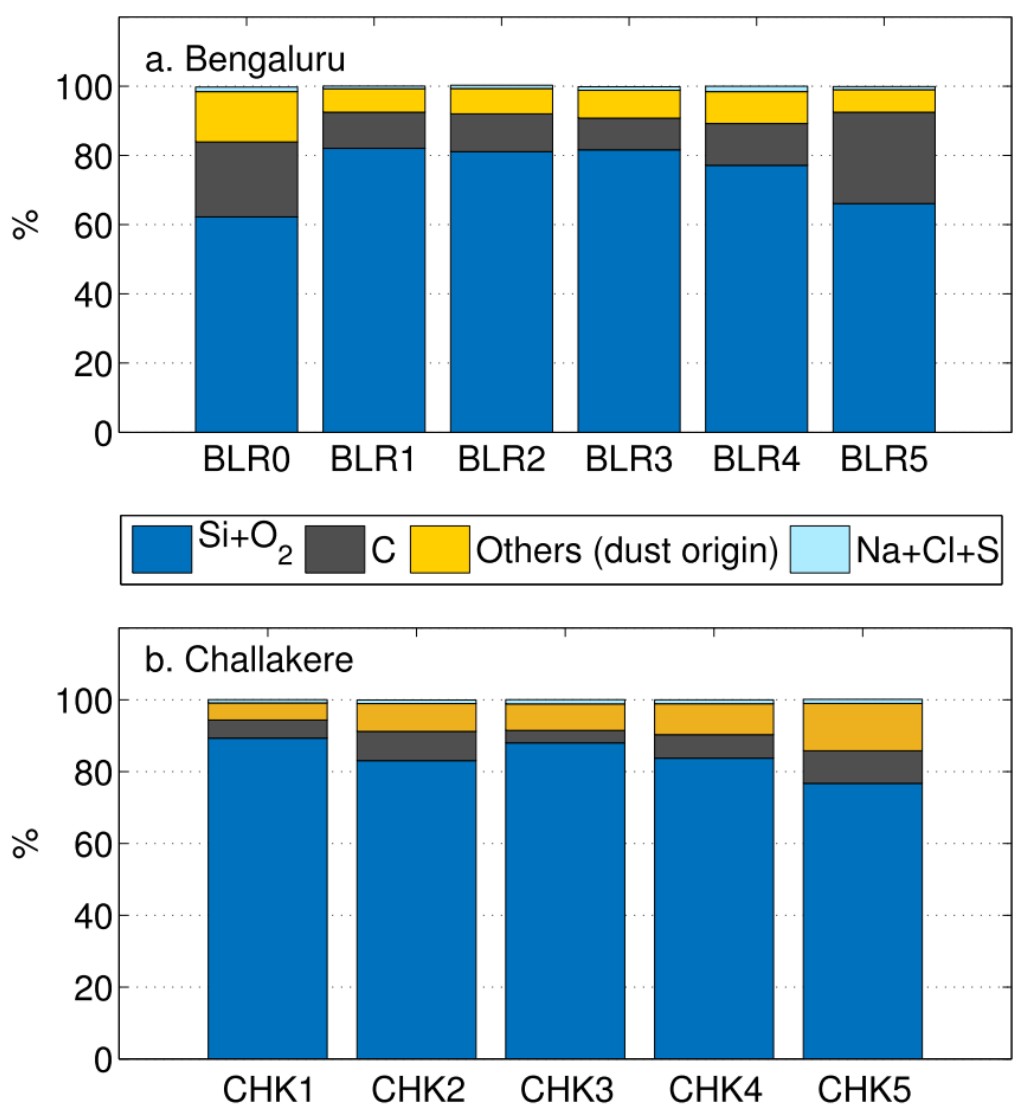

**Figure 7: Sample-wise mean composition of the collected particles for a) Bengaluru and b) Challakere.**





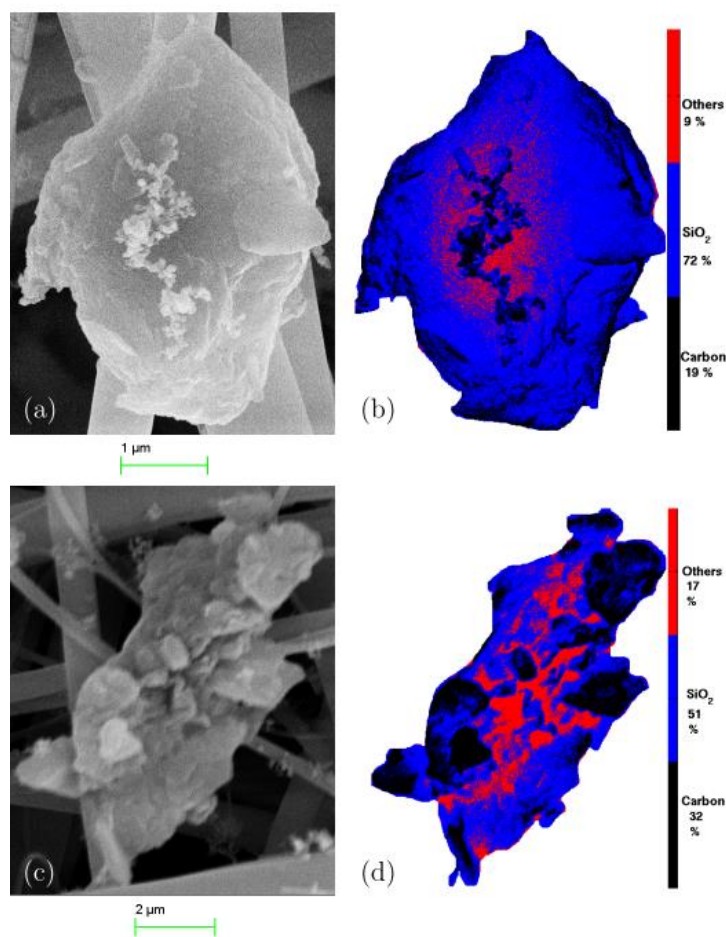

**Figure 8: a) and c) original SEM image of the selected composite particles; b) and d) composition map of the selected**
5 **particles derived using SEM images and the EDX results.**



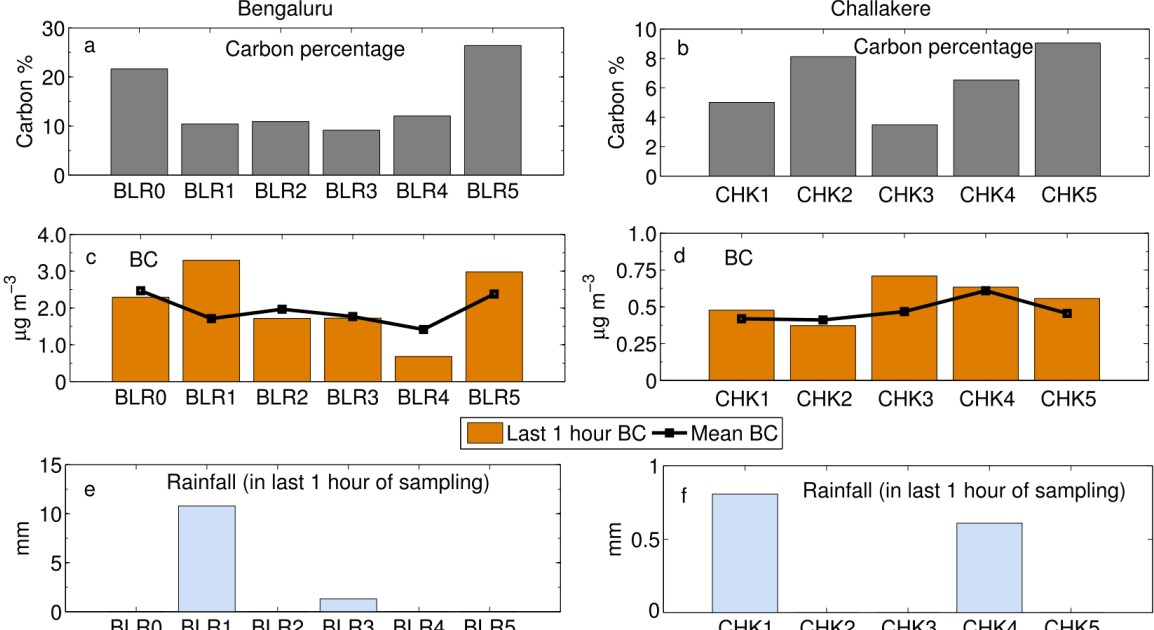

**Figure 9: Sample-wise mean carbon percentage in the collected composite particles for a) Bengaluru and b) Challakere. Measured BC mass concentration during last 1 hour of sampling for c) Bengaluru and d) Challakere. Measured rainfall during last 1 hour of sampling over e) Bengaluru and f) Challakere.**



**Figure 10: Relationship between Carbon percentage in the collected composite particles and the measured BC mass concentration for Bengaluru and Challakere samples. The color of the points indicates rainfall amount (in mm) for the last hour of sampling.**

