# Peer review of "Morphology, Chemical Composition and Mixing State of Atmospheric Aerosols from Two Contrasting Environments in Southern India"

_Atmospheric Chemistry and Physics, 2018_

## Referee Comment (RC1) · Anonymous Referee #1 · 15 Sep 2018

Morphology, Chemical Composition and Mixing State of Atmospheric Aerosols from Two Contrasting Environments in Southern India

Hariram

The authors used SEM/EDX to study aerosol particles in urban and rural sites in southern India. Basically, the aerosol information is limited from India atmosphere. I really expect that I could get some useful information from this study. However, I am disappointed about the study. The study didn't select right sample collection to do right analysis.

Firstly, the authors used quartz filters to collect aerosol particles and study individual

particles. The information could not descript the right single particles in the air. The quartz filter looks like fibrous. Many fine particles are hiding in the filter and hence the SEM could not see all the particles in the filter. I would like to say that you need to use the flat substrate such as TEM grids, silicon substrate, and polycarbonate filter. I think that the authors should look at these references cited in the paper.

Secondly, the author totally made a mistake about the core-shell structure shown in Figure 2. Core should be in center of particles and shell look like coating on the shell. In the Figure 2, the red part only overlapped on other part. I am pretty sure that the particle is not core-shell structure. Based on these two points, the rest figures could not be right data analysis. Even in Figure 4, I could not find any core-shell structure.

Thirdly, because the authors selected one wrong sample filter, they could not get any good images to secondary particles, soot (BC), metal, and other particles. EDX did show quit high Si from quartz filter. The EDS data could not be quantitatively analyzed to show C, O, and Si.

Based on these points, the paper doesn't supply any useful information about aerosol information at two contrast sites.

Other comments:

1) P5Line 4-5 how do you know the BC?

2) P5Line 33 why are Si and O dominant in the particles? You might analyze coarse particles because the fine particles penetrate in the filter. Or the quartz filter influence the EDS.

---

## Referee Comment (RC2) · Anonymous Referee #2 · 9 Oct 2018

Review of the manuscript entitled " Morphology, Chemical Composition and Mixing State of Atmospheric Aerosols from Two Contrasting Environments in Southern India" by Hariram et al., explored the possibility of analyzing the aerosol particles collected from two contrasting urban and rural sites in southern India for the mixing state using SEM-EDX. Although the results presented to be of some importance in terms of reducing the current uncertainties on the mixing state of aerosols while addressing their radiative forcing, authors need to clarify some important issues in this manuscript before it is reviewed again.

What is the significance of southern India in terms of radiative forcing of dust and black

carbon compared to heavily polluted Indo-Gangetic Plain? What is the rationale of choosing these two sampling sites and how is relevant to extrapolate to other geographical locations in southern India? Some sentences are very hard to follow. I encourage authors to clearly state their objectives and overall implications in the manuscript. Why high volume air sampling is used for this study particularly when focussing on "mixing state"? How about the sampling artefacts when discussing about the mixing state of aerosol particles as authors collected ambient aerosols from two contrasting urban and rural sampling sites in southern India on quartz fiber filters using high volume air sampler? When the ambient air is sucked, it can also be possible that on filter substrates the organic compounds and black carbon are externally/internally mixed and/or coated with other constituents of aerosols. Besides, the less number of samples collected (6 from each site) and unclear method description of within lab processing/handling of the filters as well as no section of quality control of methods presented here, I am not fully convinced that whether these results can be useful for further interpreting on the mixing state of ambient aerosols (dust/BC).

While SEM-EDX is used here to examine the mixing state of aerosols from the above two sampling sites, but the filter substrates used for this analysis is not appropriate for this kind of study. I wonder, why do authors choose quartz fiber filters for SEM-EDX analysis other than polycarbonate or Teflon filters. This is one of my major concern about the read further on the data interpretations of the manuscript. Also, quartz filters are made up of "SiO2" and traces of other elements like Na, Al. Therefore, when I looked at the outcomes of this research study, they listed that SiO2 is the base or core of the particles examined here and that too attributed to mineral dust origin. I wonder whether this SiO2 is from filter material or collected particles? Isn't this obvious that rural samples have relatively more enriched in dust and contain less carbonaceous components than urban samples? My another major concern is why the sampler was operated about 1.5m above the ground level (see page 2, line 33)? This means that the samples are not representative of even the receptor sites they have studied? How can authors' attribute such ground level collection of aerosols to the entire respective rural

and urban atmosphere? Why only monsoon season is targetted here in this study?

Line 13-15, page 2: Unclear, rewrite. Line 31, page 2: replace "far subdued" with "minor contributions from"

Overall, what is the major outcome of this study'? Conclusions: 1) Particles of different shapes present in the ambient air.

Line 13: Why the choice of only these two sites for examining the mixing state of aerosols other than the hotspot of air pollution from India? Why only monsoon season is so important other than winter? Lind 34: Why samples were collected only during the summer monsoon season? P 2, L 35: I could not follow what authors' trying to tell about? sampling duration is 6-8 hrs but the entire sampling activity is 2-3 days? Please rewrite. P3 L 1: Separate samples collected for polluted hours of morning and evening also collected clean samples in the afternoon. This distinction seems rather vague. Out of six from Bengaluru, how many of them collected like this? pl. specify here within parenthesis. P3 L4: Is this authors first time used this kind of equipment and method of study as stated in the abstract? If so, it is appropriate to give the actual details of how each sample has been prepared for the microscopic analysis on SEM-EDX rather than just pointing to somebody's reference. I encourage authors to provide the necessary details that underwent in the sample preparation for SEM-EDX analysis. Also, state what kind of quality assurance is maintained to have confidence in the results presented here? Do authors run or check some intl' standards here. Mention here all how the blanks were taken during the collection? How different they are from the real ambient samples? Without all these details, the citations to other people work are not enough. While going through the refs' cited in the MS, using the quartz fiber filters the main disadvantage is "Filters made of fibers have one main disadvantage: particulates can settle in the depths of the filter and in the mesh of the fibers (Sielicki et al., 2007)". In this review article, also stated clearly that the best substrate for SEM analysis is the polycarbonate filter. Therefore, I wonder whether this study made on quartz filters is meaning to interpret further on what the results pointed to and what

implications they have for the accurate assessment of mixing state? One image from Sielicki et al., 2007, showing the disadvantage of the quartz microfiber filters for the SEM analysis.

Cynosure? Replace with another word. L 30, P5: carbon coating percentage is estimated by excluding the carbonates? How this is done? L 10, P 7: BLR1 was collected during rainfall? Is the BLR1 collected during a rain event? In which case, how this sample is dried and analyzed on SEM-EDX.

Please also note the supplement to this comment:
https://www.atmos-chem-phys-discuss.net/acp-2018-745/acp-2018-745-RC2-supplement.pdf

---

## Referee Comment (RC3) · Anonymous Referee #3 · 14 Oct 2018

In p.2 L 34 and 35 two types of measurement system units are presented, one of the metric decimal, when writing the sampler was operated about 1.5 m above ground level, and another when mentioning the flow as at a flow rate of 40 Cubic Feet per Minute (CFM), which refers to the English system. So one or the other should be used,

It does not present information on how to prepare the samples for analysis in the SEM and EDX. Furthermore, it does not indicate each time the sampling was carried out. It does not explain why it was chosen to place the sampler at a height of 1.5 m, what would be the advantages of this placement.

ÂăThey speak of the Aethalometer which is not mentioned in the summary, nor does

it indicate the use that was given in the course of the work nor the data obtained by the use of it; there is no information on the theoretical aspects of the body of work, it is not explained in what other research has been used, as the advantages offered by its use in this type of research, nor are the aspects of its use in the section described. of methodology

---

## Author Comment (AC1) · 1 Jan 2019

We thank the reviewer for the comments and overall summary evaluation.

General comment:

1. Firstly, the authors used quartz filters to collect aerosol particles and study individual particles. The information could not descript the right single particles in the air. The quartz filter looks like fibrous. Many fine particles are hiding in the filter and hence the SEM could not see all the particles in the filter. I would like to say that you need to use the flat substrate such as TEM grids, silicon substrate, and polycarbonate filter. I think

that the authors should look at these references cited in the paper.

- This is an important point, and we agree with the reviewer that many fine particles could hide in the filter (in depths) and thus one may not see all the particles. The reviewer suggests for the use of a filter substrate with smooth surface for optimal analysis of individual particles. While we agree with this suggestion, we feel that these arguments are correct in case of a general aerosol-focussed study especially concerned with fine mode particles. We, on the other hand, are interested to study the morphology and state of mixing of dust-BC composite aerosol system (dust particles with BC depositions), which mainly falls under coarse mode aerosols. A recent study (Beatriz et al. 2016) has reported that quartz fiber filters efficiently collect particles of different size ranges in the top layer ranging from diameter of 0.199 $\mu$m to 10.272 $\mu$m. It is also reported that the largest volume fraction (34.216 %) of particles captured is in the outer layer. Moreover, it is also estimated that about 65 % of the volume fraction of particles is found in the top two layers of the filter (Beatriz et al. 2016). Additionally, it is reported that, the inner layers are found to trap only the fine particles of mean particle diameter 0.533 $\mu$m. In this study, since our interest was centered around the coarse mode aerosols, we could easily identify them from the top layer of the quartz filter. Moreover, use of aerosol loaded quartz filter for automated SEM/EDX studies, has also been supported in the literature (e.g., Willis et al., 2003) Our study was completed with a semi-automated SEM/EDX system and a careful post processing with the help of an image processing software which solved the tribulations of the noisy fibrous background. The software helped in enhancing the image and cropping the independent particles easily from background. Thus, with the help of sophisticated instrumentation and post-processing, we examine the state of mixing of dust-BC aerosol system over two stations with contrasting environments in southern India.

2. Secondly, the author totally made a mistake about the core-shell structure shown in Figure 2. Core should be in center of particles and shell look like coating on the shell. In the Figure 2, the red part only overlapped on other part. I am pretty sure that the

particle is not core-shell structure. Based on these two points, the rest figures could not be right data analysis. Even in Figure 4, I could not find any core-shell structure.

- We agree with the reviewer that core should be in the center of the particle and the shell should form a covering on it, in a totally core-shell structure. However, in this study we find particles to be only partially coated and some regions of the core part of the particles appear to be exposed without any shell cover on it and this is one of the important finding. Such partially coated particles when viewed from top, look similar to the ones shown in figure 2. Thus, on the account of the projection of the image, the partially coated particles reported in this study, do not look strictly similar to the theoretical shell-core structured particles. Some of the particles reported in figure 4 also depict such partially coated shell-core structure. Small particles that seem to adhere on the coarse particle like BC chain found on a portion of the surface of a bigger particle as in figure-8 is also considered as coating.

3. Thirdly, because the authors selected one wrong sample filter, they could not get any good images to secondary particles, soot (BC), metal, and other particles. EDX did show quit high Si from quartz filter. The EDS data could not be quantitatively analyzed to show C, O, and Si.

- We would like to undeniably state that all the chemistry data reported in the paper is qualitative. With the present-day available techniques, it is impossible to probe into the individual particle level chemistry on a quantitative basis. As such, we have used the energy dispersive spectroscopy (EDX) which is a widely used qualitative approach to examine the elemental composition of atmospheric aerosols. In this study, we have performed multiple EDX analysis on different portions of the same particle to arrive at the average elemental composition to minimize the error. Also, the concern of background artefact was studied by performing an experiment comparing with the aerosol samples collected on a carbon tape. We computed the background error of 2.05±0.8 % together contributed by Si & O2. About 50 independent coarse mode particles were used for this study. Furthermore, the EDX was performed carefully on clearly exposed

surface of the particles with minimum possibility of background interference. This increases our confidence in the reported results. These are included in the revised ms.

Other comments:

1) P5Line 4-5 how do you know the BC?

- We have used an Aethalometer (model AE42; Magee Scientific, USA) for simultaneous measurements of ambient black carbon (BC) mass concentration. Those measurements have been utilised in this study to examine their association with carbon coating on dust particles. Details are provided in the revised ms

2) P5Line 33 why are Si and O dominant in the particles? You might analyse coarse particles because the fine particles penetrate in the filter. Or the quartz filter influence the EDS.

- As already mentioned in reply to 1st general comment from the reviewer, the focus of our work is coarse mode aerosol particles. These particles could be easily identified with sufficiently high magnification (50,000-1,00,000) under SEM as they mostly get trapped in the outer most layer of the quartz filter, unlike the fine mode particles which penetrate the filter. Since the coarse mode particles are mainly composed of Si and O (silica i.e. dust origin), we get dominance of these elements in EDS. The concern of background artefact (quartz filter) in EDS results has been analysed by performing an experiment comparing with the aerosol samples collected on a carbon tape. We have computed the background error of 2.05±0.8 % together contributed by Si & O. About 50 independent coarse mode particles were used for this study. Also EDX was performed carefully on clearly exposed surface of the particles with minimum possibility of background interference. This increases our confidence in the reported results.

References

Beatriz, S.-P.; Luis, N.; Leonor, C.; Laura, M.; Elena, M.; Yolanda, F.-N. Imaging Techniques and Scanning Electron Microscopy as Tools for Characterizing a Si-Based Material Used in Air Monitoring Applications. Materials 2016, 9, 109. Willis, R. D. AND T L. Conner. GUIDELINES FOR THE APPLICATION OF SEM/EDX ANALYTICAL TECHNIQUES FOR FINE AND COARSE PM SAMPLES. U.S. Environmental Protection Agency, Washington, DC, EPA/600/R-02/070 (NTIS PB2004-100988), 2003.

---

## Author Comment (AC2) · 1 Jan 2019

We thank the reviewer for the comments and overall summary evaluation.

1) What is the significance of southern India in terms of radiative forcing of dust and black carbon compared to heavily polluted Indo-Gangetic Plain?

The present study aims to identify the state of mixing of dust and BC in two relatively low dust-laden environments of southern India; one a remote semi-arid location and the other an urban location not very far from the former. Mixing state of dust and BC in such contrasting environments have been less explored compared to the heavily

dust-laden Indo-Gangetic Plain. Such a study between two distinctively differing, yet nearby environments would be important in delineating the associated differences in the aerosol-system (specifically dust-BC) over these stations, especially their state of mixing. Furthermore no reliable data on the mixing state of aerosols are available from these regions (unlike the Indo-Gangetic Plains, which are more explored (Jethva et al.,2005; Ramanathan and Ramana 2005; Lau et al., 2006; Gautam et al., 2011)) and often has to be assumed in various modelling studies. Another important aspect of the study is exploring the effect of rainfall on the mixing morphology of dust-BC, as well as the dependency of the mixing state on ambient BC mass concentration. Considering the importance to the regional climate, the aerosol properties over the Indo-Gangetic Plain has been widely explored.

2) What is the rationale of choosing these two sampling sites and how is relevant to extrapolate to other geographical locations in southern India?

We chose two sampling sites namely Bengaluru and Challakere, which to an extent would represent ambient conditions over a polluted urban centre and a region almost completely free from any anthropogenic activity (Satheesh et al., 2013) respectively. We wanted to study the state of mixing of dust and BC over such contrasting environments in terms of local BC and dust burden. The urban station Bengaluru, on account of its high emissions (from industries and the huge population of automobiles) would have more anthropogenic aerosol loading (including BC), while Challakere would naturally have more dust burden due to surrounding barren land unlike the densely populated landscape of Bengaluru. Thus, this comparison offers us an excellent opportunity to examine the state of mixing of dust-BC system in two contrasting environments. Thus, we chose two such sampling sites. Though the sites represent ambient conditions over an urban and a rural station. We do acknowledge that the local conditions over any other site in southern part of India would vary depending upon local meteorology, land-scape, geographical location and topography and hence do not attempt to extrapolate the observed state of mixing of dust-BC to any other region.

3) I encourage authors to clearly state their objectives and overall implications in the manuscript.

The main objectives of the present study are as follows: Examine the state of mixing of dust and BC aerosol species in two contrasting environments – one with low local anthropogenic emissions and one with very high anthropogenic emissions and see how the interplay between dust and BC can modulate the coating percentage. Additionally, we report how precipitation impacts the mixing state of dust-BC aerosol system. The above aspects are now clearly stated in the revised ms.

4) Why high volume air sampling is used for this study particularly when focussing on mixing state?

In this study our main objective was to understand the mixing of dust and BC and we have not focussed on all the aerosol species. Hence, we decided to leave out volatile species and smaller particles and concentrate on dust and BC. We felt that a high volume sampler would be adequate for our purpose (Beatriz et al. 2016) as we wanted more samples that could be used for different analysis. More details on the facts that led us to use a high volume with quartz filter are given in the reply to next question (question no. 6). After analysing the pros and cons we have combined a low volume sampler (using PTFE filters) along with the high volume sampler in our subsequent studies; but the results are not included in this study.

5) How about the sampling artefacts when discussing about the mixing state of aerosol particles as authors collected ambient aerosols from two contrasting urban and rural sampling sites in southern India on quartz fiber filters using high volume air sampler?

When the ambient air is sucked, it can also be possible that on filter substrates the organic compounds and black carbon are externally/internally mixed and/or coated with other constituents of aerosols. Besides, the less number of samples collected (6 from each site) and unclear method description of within lab processing/handling of the filters as well as no section of quality control of methods presented here, I am not

fully convinced that whether these results can be useful for further interpreting on the mixing state of ambient aerosols (dust/BC).

The most important comment raised by the reviewers in common is regarding the aerosol sampling using the quartz fiber filter for scanning electron microscopic study. Reviewer #1 mentioned that the probability of fine particles to hide within the porous filter cannot be neglected. In such a case, those fine particles may not be captured by the microscope, for which Reviewer #1 suggests for the usage of a filter substrate with smooth surface for optimal analysis of individual particles. The Reviewer #2 also raised the similar concern pointing out the reference from Sielicki et al. 2011, that the quartz filters are not the perfect choice for scanning electron microscopic studies as many particles will be settled in the depths of the filter. These arguments are correct to a large extent in case of a general study spanning aerosol sizes over several orders of magnitude, including the ultra-fine and coarse mode particles. However, Beatriz et al. 2016 has reported that quartz fiber filters efficiently collect particles of different size ranges in the top layer ranging from diameter of 0.199 $\mu$m to 10.272 $\mu$m. The study also tells that the largest volume fraction (34.216 %) of particles captured is in the outer layer. Moreover, the study has estimated that about 65 % of the volume fraction of particles was found in the top two layers of the filter. As mentioned by the reviewers, the inner layers were found to trap only the fine particles of mean particle diameter 0.533 $\mu$m. But our interest of study is mainly dust particles and its state of mixing which falls in the coarse mode size range. So, it was easy to pick them from the top layers of the quartz filter. Also, it is reported by Willis et al. 2003 that quartz filters with aerosol samples can be considered for certain types of SEM/EDX studies which are supported with automation. Our study was completed with a semi-automated SEM/EDX system and a careful post processing with the help of an image processing software which solved the tribulations of the noisy fibrous background. The software helped in enhancing the image and cropping the independent particles easily from background. This is illustrated in Fig-2. The method of sample preparation is given with the response of Comment-13. These aspects are incorporated in the revised ms.

6) About sampling. Reviewer comments that use of Quartz filter paper is not suitable for microscopic study and the resultant chemistry has contributions from filter paper also.

All the chemistry data reported in the paper is qualitative. With the present-day available techniques, it is impossible to probe into the individual particle level chemistry on a quantitative basis. So, we have used the energy dispersive spectroscopy (EDX), which is a widely used qualitative approach to examine the elemental composition of atmospheric aerosols. In this study, we have performed multiple EDX analysis on different portions of the same particle to arrive at the average elemental composition to minimise the error. Also, the concern of background artefact was studied by performing an experiment comparing with the aerosol samples collected on a carbon tape. We computed the background error of 2.05±0.8 % together contributed by Si and O2. About 50 independent coarse mode particles were used for this study. Although more particles should have been examined to get a statistically sound estimate, the practical constraints forced us to limit to 50 particles. Also, EDX was performed carefully on clearly exposed surface of the particles with minimum possibility of background interference. While doing this we have taken utmost care to choose bigger particles which are clearly exposed which leaves out the possibility of mixing up the Si background and the Si in the particles. This ensures minimum errors introduced by background artefact and strengthens the confidence in the observed values. These are clearly stated in the revised ms

7) Isn't this obvious that rural samples have relatively more enriched in dust and contain less carbonaceous components than urban samples?

We agree with the reviewer's comments that rural site will contain less carbonaceous matter; however, with regard to dust, it need not be necessarily so. So, our study brings out the different aspects of the state of mixing depending on the ambient BC concentration as well as local meteorology. As mentioned earlier this study has been carried out as part of a bigger effort to understand the dust-BC mixing over the Indian

region. Nevertheless, an appropriate correction has been made in the manuscript.

8) My another major concern is why the sampler was operated about 1.5 m above the ground level (see page 2, line 33)? This means that the samples are not representative of even the receptor sites they have studied? How can authors attribute such ground level collection of aerosols to the entire respective rural and urban atmosphere? Why only monsoon season is targeted here in this study?

The main constraint in the sample collection in our urban location was the availability of open space. Thus, in order to avoid the influence of the orography we had to carry out the sampling from the rooftop of a building at a height ∼15 m above the surface. The height was mentioned as 1.5 m in the ms and has been corrected in the modified version. The sampling site has proper rain protection (in addition to the sampler rain shield) and the obstruction due to the orography was negligible.

9) Why the choice of only these two sites for examining the mixing state of aerosols other than the hotspot of air pollution from India?

As mentioned in the authors' reply to the 1st comment, we would like to examine the state of dust-BC mixing in different environments the in the future.

10) Why only monsoon season is so important other than winter? Line 34: Why samples were collected only during the summer monsoon season?

The samples were collected as part of an effort to collect data on the mixing state. We are in the process of collecting continuous and extensive data from various sites. This is just the initial effort in this direction. The data availability during the monsoon season is scarce compared to the other seasons (especially the AOD, extinction profiles and the type of aerosol) and hence we examined the dust-BC mixing state during this period. We agree with the reviewer that, sampling in the winter season would give a chance to sample more number of particles. However, the summer monsoon season provides an excellent environment to examine the state of mixing of dust-BC system

when the prevalent large-scale meteorological conditions bring in a possibility of long range transport of dust (Lau et al., 2006, Begum et al., 2011; Govardhan et al., 2015) and influence of rain on the aerosol mixing state. This would also allow us to examine the background aerosol mixing state during periods following extended rainfall events.

11) P 2, L 35: I could not follow what authors trying to tell about? sampling duration is 6-8 hrs but the entire sampling activity is 2-3 days? Please rewrite.

The sentence has been modified in the revised version of the manuscript.

12) P3 L 1: Separate samples collected for polluted hours of morning and evening also collected clean samples in the afternoon. This distinction seems rather vague. Out of six from Bengaluru, how many of them collected like this? pl. specify here within parenthesis.

The aforementioned details about sampling are added in the modified version of the manuscript.

13) P3 L4: Is this authors first time used this kind of equipment and method of study as stated in the abstract? If so, it is appropriate to give the actual details of how each sample has been prepared for the microscopic analysis on SEMEDX rather than just pointing to somebody's reference. I encourage authors to provide the necessary details that underwent in the sample preparation for SEM-EDX analysis. Also, state what kind of quality assurance is maintained to have confidence in the results presented here? Do authors run or check some intl standards here. Mention here all how the blanks were taken during the collection? How different they are from the real ambient samples? Without all these details, the citations to other people work are not enough.

We comply with this suggestion and the details are now provided in the revised ms

The filter loading and the process of sampling were done following the usual protocols. All operations were done with gloves and forceps to ensure that no external contamination has happened. The collected samples were stored in an airtight desiccator with

silica gel. The sample loaded filter papers were then prepared for SEM analysis by shredding into squares of area ∼1 cm2. These samples were then carefully adhered to aluminium stubs using conductive carbon tape and were lined with conductive silver paste. In order to avoid charging tribulations and to make the sample conductive, a very thin film (∼15 nm) of Gold (Au) was overlaid on the surface of the samples using a sputter coater under Argon gas atmosphere. Similar samples were prepared for SEM/EDX analysis from different portions of each sampled filter paper. All operations were done in the clean room facility of Micro and Nano Characterization Facility (MNCF) located at CeNSE, IISc, Bengaluru, funded by NPMAS-DRDO and MCIT, MeitY, Government of India following proper standards and regulations. We have not done any quality assurance tests and no quantitative study regarding the chemical composition. Quality assurance was done in the operations, sample collection and preparation ensuring the possibility of minimum errors and contamination. We have performed one experimental study to analyse the background influence of Quartz filter paper in the reported EDX chemistry and is mentioned in response-6. As discussed in the above comment regarding chemistry (response-6), all chemistry details reported in this paper are qualitative.

14) While going through the refs cited in the MS, using the quartz fiber filters the main disadvantage is Filters made of fibers have one main disadvantage: particulates can settle in the depths of the filter and in the mesh of the fibers (Sielicki et al., 2007). In this review article, also stated clearly that the best substrate for SEM analysis is the polycarbonate filter. Therefore, I wonder whether this study made on quartz filters is meaning to interpret further on what the results pointed to and what implications they have for the accurate assessment of mixing state? One image from Sielicki et al., 2007, showing the disadvantage of the quartz microfiber filters for the SEM analysis.

The same issue was raised by Reviewer #1 and we have already answered this in our response to question no 5.

15) Carbon coating percentage is estimated by excluding the carbonates? How this is

done?

We examined the total elemental percentage of independent EDX results. This gives us a general idea of the elements present in the particles. The possibility of Silicon carbide and Aluminium carbonate are most unlikely. About 10% of the observed particles had relatively higher percentage of elements that can form carbonates (Calcium, Magnesium, Sodium and Potassium). Our interest was to see the percentage of black carbon which is independent of the carbon percentage from carbonates and so we excluded them from the calculation. Similarly, the presence of BC was often verified manually with its distinct chain type morphology as shown in figure-7. So the carbon reported in this study is the percentage of black carbon alone.

16) BLR1 was collected during rainfall? Is the BLR1 collected during a rain event? In which case, how this sample is dried and analyzed on SEM-EDX.

The samples were not dried separately and we were operating with proper rain shields. The samples were not wet or damaged. The filter loading and the process of sampling were done following the usual protocols. All operations were done with gloves and forceps to ensure that no external contamination has happened. The collected samples were stored in an airtight desiccator with silica gel. More details are given in our response to question 13. Post processing operations were done in the clean room facility of Micro and Nano Characterization Facility (MNCF) located at CeNSE, IISc, Bengaluru, funded by NPMAS-DRDO and MCIT, MeitY, Government of India following proper standards and regulations.

References:

Ramanathan, V., and M. V. Ramana. "Persistent, widespread, and strongly absorbing haze over the Himalayan foothills and the Indo-Gangetic Plains." Pure and Applied Geophysics 162.8-9 (2005): 1609-1626.

Gautam, R., et al. "Accumulation of aerosols over the Indo-Gangetic plains and

southern slopes of the Himalayas: distribution, properties and radiative effects during the 2009 pre-monsoon season." Atmospheric Chemistry and Physics 11.24 (2011): 12841-12863.

Jethva, Hiren, S. K. Satheesh, and J. Srinivasan. "Seasonal variability of aerosols over the Indoâ ĂŘGangetic basin." Journal of Geophysical Research: Atmospheres 110.D21 (2005).

Lau, K. M., M. K. Kim, and K. M. Kim. "Asian summer monsoon anomalies induced by aerosol direct forcing: the role of the Tibetan Plateau." Climate dynamics 26.7-8 (2006): 855-864.

Begum, Bilkis A., et al. "Long–range transport of soil dust and smoke pollution in the South Asian region." Atmospheric Pollution Research 2.2 (2011): 151-157.

Govardhan, Gaurav, et al. "Performance of WRF-Chem over Indian region: Comparison with measurements." Journal of Earth System Science 124.4 (2015): 875-896.

Beatriz, S.-P.; Luis, N.; Leonor, C.; Laura, M.; Elena, M.; Yolanda, F.-N. Imaging Techniques and Scanning Electron Microscopy as Tools for Characterizing a Si-Based Material Used in Air Monitoring Applications. Materials 2016, 9, 109.

Sielicki, Przemysław, et al. "The progress in electron microscopy studies of particulate matters to be used as a standard monitoring method for air dust pollution." Critical reviews in analytical chemistry 41.4 (2011): 314-334.

Satheesh, S. K.; Moorthy, Krishna K.; Srinivasan, J. (2013) New directions: elevated layers of anthropogenic aerosols aggravate stratospheric ozone loss? Atmospheric Environment, 79. pp. 879-882. ISSN 1352-2310

Willis, R. D. and T L. Conner. GUIDELINES FOR THE APPLICATION OF SEM/EDX ANALYTICAL TECHNIQUES FOR FINE AND COARSE PM SAMPLES. U.S. Environmental Protection Agency, Washington, DC, EPA/600/R-02/070 (NTIS PB2004-100988), 2003.

---

## Author Comment (AC3) · 1 Jan 2019

We thank the reviewer for the comments and overall summary evaluation.

1) In p.2 L 34 and 35 two types of measurement system units are presented, one of the metric decimal, when writing the sampler was operated about 1.5 m above ground level, and another when mentioning the flow as at a flow rate of 40 Cubic Feet per Minute (CFM), which refers to the English system. So one or the other should be used

The units are corrected, as suggested

2) It does not present information on how to prepare the samples for analysis in the

SEM and EDX.

The filter loading and the process of sampling were done following the usual protocols. All operations were done with gloves and forceps to ensure that no external contamination has happened. The collected samples were stored in ziplock covers and stored in airtight desiccator.

The sample loaded filter papers were then prepared for SEM analysis by shredding into squares of area ~1 cm2. These samples were then carefully adhered to aluminium stubs using conductive carbon tape and were lined with conductive silver paste. In order to avoid charging tribulations and to make the sample conductive, a very thin film (~15 nm) of Gold (Au) was overlaid on the surface of the samples using a sputter coater under Argon gas atmosphere. Similar samples were prepared for SEM/EDX analysis from different portions of each sampled filter paper.

All operations were done in the clean room facility of Micro and Nano Characterization Facility (MNCF) located at CeNSE, IISc, Bengaluru, funded by NPMAS-DRDO and MCIT, MeitY, Government of India following proper standards and regulations. These changes are incorporated in the revised manuscript.

3) Furthermore, it does not indicate each time the sampling was carried out. The details were included only in the table.

Sample name – Place – Date – Start time-End time (IST hours) – No. of particles examined

BLR0 – Bengaluru – 08-08-16 – 18:00-23:00 – 21

BLR1 – Bengaluru – 19-08-17 – 09.45-16.45 – 41

BLR2 – Bengaluru – 19-08-17 – 22.30-06.30 – 47

BLR3 – Bengaluru – 20-08-17 – 09.30-15.30 – 59

BLR4 – Bengaluru – 20-08-17 – 19:00-03:00 – 37

BLR5 – Bengaluru – 30-08-17 – 18:00-02:00 – 44

CHK1 – Challakere – 16-08-17 – 14.00-19.00 – 34

CHK2 – Challakere – 17-08-17 – 00:30-06:30 – 24

CHK3 – Challakere – 17-08-17 – 09:00-15:00 – 26

CHK4 – Challakere – 17-08-17 – 18.00-23.30 – 31

CHK5 – Challakere – 18-08-17 – 03:00-10:00 – 36

Details (date, duration and number of particles analysed) about sampling carried out at Bengaluru (BLR) and Challakere (CHK). The samples BLR1, BLR3, CHK1 and CHK3 are collected during the cleaner hours of afternoon, while the remaining ones are collected during the relatively polluted hours. The aforementioned details about sampling are added in the revised version of the manuscript.

4) It does not explain why it was chosen to place the sampler at a height of 1.5 m, what would be the advantages of this placement.

The main constraint in the sample collection in our urban location was the availability of open space. Thus, in order to avoid the influence of the orography we had to carry out the sampling from the rooftop of a building at a height ∼15 m above the surface. The height was mentioned as 1.5 m in the ms and has been corrected in the modified version. The sampling site has proper rain protection (in addition to the sampler rain shield) and the obstruction due to the orography was negligible.

5) They speak of the Aethalometer which is not mentioned in the summary, nor does it indicate the use that was given in the course of the work nor the data obtained by the use of it; there is no information on the theoretical aspects of the body of work, it is not explained in what other research has been used, as the advantages offered by its use in this type of research, nor are the aspects of its use in the section described of methodology.

It has been pointed out by other reviewers too. We have added a short section in the revised manuscript. We have used the aethalometer data mainly to measure the mass concentration of BC in the atmosphere. It will tell us whether the amount of BC itself is changing or not and provide a scale to determine whether the changes in the composition is directly associated with the changes in BC concentration. But this cannot be used as a quantitative scale for the changes in the composition as we are trying to understand the coating of BC over other particles mainly dust.